# Location of Trigger Points in a Group of Police Working Dogs: A Preliminary Study

**DOI:** 10.3390/ani13182836

**Published:** 2023-09-07

**Authors:** Maira Rezende Formenton, Karine Portier, Beatriz Ribeiro Gaspar, Lisa Gauthier, Lin Tchia Yeng, Denise Tabacchi Fantoni

**Affiliations:** 1School of Veterinary Medicine and Animal Science, University of São Paulo, São Paulo 05508-270, Brazil; beatriz.gaspar@usp.br (B.R.G.); dfantoni@usp.br (D.T.F.); 2VetAgro Sup (Campus Vétérinaire), Centre de Recherche et de Formation en Algologie Comparée (CREFAC), University of Lyon, 69280 Marcy l’Etoile, France; karine.portier@vetagro-sup.fr (K.P.); lisagauthier97@gmail.com (L.G.); 3Centre National de la Recherche Scientifique (CNRS), Institut National de la Santé et de la Recherche Médicale (INSERM), Centre de Recherche en Neurosciences de Lyon (CRNL), University of Lyon, U1028 UMR 5292, Trajectoires, 69500 Bron, France; 4School of Medicine, Institute of Orthopedics and Traumatology, University of São Paulo, São Paulo 05403-010, Brazil; lintyeng@uol.com.br

**Keywords:** myofascial pain, analgesia, muscle pain, myofascial pain syndrome, athletic dogs

## Abstract

**Simple Summary:**

This research looked at the number and position of sore points in police working dogs. The study selected twelve dogs from a military police kennel based on convenience. Only dogs that were active, had no other health issues or changes seen in X-rays, and were involved in six hours of intense physical activity each day were included. The dogs underwent examinations to check their health, and two independent examiners inspected them to find any sore points called trigger points (TPs). The location of the TPs was noted using an anatomical figure. The highest percentage of TPs was found in the lower back muscles (42%), followed by the muscles in the back, groin, thigh, and inner thigh (33%). Most of the TPs were found on the right side of the body. This study found that police working dogs had a higher percentage of TPs in their spinal and hind limb muscles, especially on the right side. These findings can help improve methods to prevent muscle pain and reduce the need for early retirement due to musculoskeletal pain in these dogs. It also brings attention to this problem that can affect dogs.

**Abstract:**

This study examined the percentage and location of trigger points in police working dogs. Twelve dogs housed at a military police kennel were selected through convenience sampling. Only active dogs with no comorbidities or radiographic changes doing 6 hours of intense physical activity per day were included. After orthopedic and neurological examination, dogs were palpated for the detection of trigger points (TPs), carried out by two independent examiners, with criteria of palpations previously standardized. TPs were recorded using an anatomy reference image according to the corresponding anatomical location. The percentage of TPs was highest in the lumbar portion of the *longissimus dorsi* muscle (42%), followed by the *latissimus dorsi*, *pectineus*, *quadriceps femoris*, and *sartorius* (33%) muscles. Most TPs were located on the right side of the body. This study’s percentage of TPs in police working dogs was higher in spinal and hind limb muscles, especially on the right side. The major criteria for identifying TPs in dogs were the pain responses to palpation and contractile local response. The findings of this study could be used to refine myofascial pain prevention to reduce early retirement due to musculoskeletal pain and draw attention to this kind of problem that can also affect dogs.

## 1. Introduction

Myofascial pain syndrome (MPS) is a musculoskeletal disorder affecting muscles, fasciae, and ligament attachments [1]. Pain is one of the cardinal signs of this condition, along with local motor and autonomic changes [2,3]. Trigger points (TPs) are nodules that elicit pain and a characteristic motor response (contractile reflex) to manual palpation or delicate needle stimulation [4,5] and are a pathognomonic sign of MPS. In humans, painful responses associated with TPs are described as a deep burning sensation, which is challenging to locate and radiates to other areas [6,7]. 

The prevalence of MPS in small animals has not been determined. However, the condition is thought to be common and underdiagnosed due to a lack of understanding among veterinary practitioners [8,9]. In animals, accurate MPS diagnosis requires specific training focused on myofascial pain recognition [5], locating taut bands and TPs, and excluding other diseases [2,3]. Even in human patients, who can verbalize pain perception and location, there is no consensus on the best diagnostic method [5,10]. Previous training [10] and complementary diagnostic methods [11,12], such as pressure algometry [13] and thermography [14,15], can be used to minimize diagnostic discrepancies.

Criteria used for TP diagnosis have not been validated in veterinary patients [16] and are often extrapolated from human studies [17]. In one of the few studies examining TP location in small animals [9], of 48 lame dogs which had not responded to pharmacological treatment, 31 dogs had TPs (82 TPs in total) and 17 had no TPs at all. Trigger points were located primarily in the *triceps brachii*, *infraspinatus*, *biceps brachii*, *pectineus*, *peroneus longus*, *gluteus medius*, *iliocostal*, and *quadriceps femoris* muscles. The fact that only 8 out of 82 TPs persisted after dry needling or infiltration emphasizes the efficacy of TP therapy when the diagnosis is correct. The findings reported in this study [9] remain to be confirmed in different scenarios and dog breeds. 

In humans, myofascial pain may lead to decreased performance and loss of function, significantly impacting mood and quality of life [18,19,20]. For example, human athletes with myofascial pain in the gastrocnemius muscle are more prone to experience symptoms of central sensitization, such as catastrophism and magnification of pain, than athletes with no TPs at that location [18]. 

Also, given that muscle fatigue and overuse are common causes of TPs, the prevalence and location of these points vary according to the physical activity performed, as shown by a study with 180 athletes from six different sports disciplines [20]. The *trapezius*, *quadratus lumborum*, *and quadriceps femoris* muscles were the most affected in judo athletes. In contrast, Taekwondo athletes tended to have more TPs in the *trapezius* and *triceps surae* muscles.

This study was designed to examine the presence and location of TPs in a group of police working dogs using palpation, establishing criteria for palpation and animals’ reaction due to it. The hypothesis was that police working dogs would have myofascial pain referred from trigger points in different muscle groups.

## 2. Materials and Methods

Working dogs selected from a military police kennel were used in this study. Guardians signed a free and informed consent form. The Institution’s Ethics Committee for the Use of Animals of the University of São Paulo approved the study protocol (No. 1571120219). 

Dogs were recruited using convenience sampling according to on-site availability. Active male and female dogs with no history of musculoskeletal injuries or lameness, weighing between 20 and 40 kg and aged 2 to 8 years, were included. Exclusion criteria were as follows: history or suspicion of neoplasia of any kind, dogs with clinical diseases or syndromes (e.g., heart, liver, kidney, or skin diseases), obese animals (body condition score 8 or 9 in a 0 to 9 scale in which “0” corresponds to cachexia and “9” to obesity; Laflamme, 1997) [21], aggressive dogs that resented being touched, dogs with neurological (e.g., peripheral neuropathies, proprioceptive or sensory deficits or seizures) or orthopedic (e.g., elbow or coxofemoral dysplasia, patellar or coxofemoral luxation, immunomediated or infectious arthritis, or cranial cruciate ligament injury) conditions detected on clinical/orthopedic examination. 

Other exclusion criteria were signs of lameness, joint crepitus, instability or edema, angular limb deformities or signs of limb rotation while walking or trotting, muscle hypotrophy or asymmetry, or other muscular changes confirmed by perimetric measurement. Dogs with signs of joint pain during clinical evaluation (regardless of radiographic findings) were also excluded. Any joints or sites (including the spine) with signs of crepitus detected in the clinical examination were radiographed to rule out subclinical musculoskeletal diseases. 

Selected dogs belonged to the same battalion, were housed in kennels of similar size, flooring, and structure, were fed the same diet, and were submitted to similar training and work schedules. Physical activities were restricted for the selected dogs, either training- or work-related, in the 24/48 h before data collection to avoid muscle pain related to the physical activity interfering with the examination. Dogs recruited for the study were primarily used for search and rescue, explosives detection, drug sniffing, or as guard dogs. 

Dogs were then submitted to palpation of major muscle groups. Muscle palpation was carried out by two independent blinded evaluators. One evaluator palpated all regions of the animal in the order described below, and afterwards, the other evaluator examined the same animal. In order to not create bias, the order of the evaluator’s examination was alternated. Myofascial pain recognition criteria and palpation methods were standardized. Muscle palpation followed the order described below. An evaluation form with muscles listed in sequence was used. 

Muscles of the head and neck, particularly the masseter, brachiocephalicus, sternocephalicus, and trapezius pars cervicalis, were palpated first. Next, front limb muscles (pectorales superficialis and pectoralis profundus), supraspinatus, infraspinatus, deltoideus, brachioradialis, extensor carpi radialis, extensor digitorum communis, extensor carpi ulnaris, flexor carpi ulnaris, triceps brachii, brachialis, biceps brachii, and omotransversarius were palpated. Then, the back muscles (trapezius pars thoracica, latissimus dorsi, longissimus thoracicae, longissimus lumborum, and serratus ventralis thoracis) were palpated. Finally, pelvic limb muscles (gluteus superficialis, gluteus medius, biceps femoris, sartorius, tensor fasciae latae, quadriceps femoris, gracilis, semitendinosus, semimembranosus, fibularis peroneus longus and brevis, flexor digitorum superficialis, flexores digitorum profundi, gastrocnemius, extensor digitorum longus, tibialis cranialis, pectineus, and iliopsoas) were palpated. The number of muscles palpated totaled 41. However, accounting for subdivisions of larger muscle groups (Figure 1), 49 muscle regions were palpated, resulting in 98 palpations/assessments per dog, including the right and left sides.

Myofascial pain diagnostic criteria consisted of palpation of a distinct hyperreactive point or nodule eliciting a typical contractile response suggestive of pain, as shown in complementary Appendix A [22]. Vocalization, turning the head towards or looking at the target area or examiner, growling or attacking, flexing the spine, or exhibiting an escape behavior during TP palpation were defined as pain responses in this study [8,17,23]. Trigger points detected by at least one examiner according to the criteria above were listed in individual evaluation forms and marked on a reference image with the corresponding anatomical position. Large muscle groups, such as the *latissimus dorsi*, were subdivided into smaller areas to facilitate accurate TP location (Figure 1). Anatomical references in Appendix B were used for the demarcation of these areas. 

Once data collection was completed, the battalion military veterinarians were trained to perform muscle stretching, to treat myofascial pain, and on prevention techniques. 

### Statistical Analysis

Dogs in this exploratory study were recruited using convenience sampling according to on-site availability. However, the minimum sample size for external validation (i.e., the prevalence of myofascial pain in working dogs overall) was estimated in advance. If we hypothesize an overall prevalence of 75%, a sample comprising 51 dogs would be required to obtain a confidence interval of 90% with a margin of error of 10%. The formula used to calculate this sample was as follows: $$n=\frac{p\left(1-p\right)z_{\gamma }^{2}}{\varepsilon^{2}}$$

In this formula, *n:* estimated sample size. *ε:* error. *p:* real percentage we wish to find. zγ: normal distribution in the quartile $\frac{1-\gamma }{2}$ Considering a 90% confidence interval: z0.9 = −1.6448. The percentage of TPs was calculated per muscle (or area in the case of large muscle groups) based on data collected by each evaluator and their intersection. Each muscle assessment was treated as an independent measure. The percentage was expressed as the number (No) of muscles affected/number of muscle groups or areas assessed per side (12 dogs).

## 3. Results

### 3.1. Animals

Sixteen dogs were initially selected. Four of these were excluded due to radiographically confirmed articular or spinal changes (two dogs and one dog, respectively) or recent history of diarrhea (one dog). The final sample comprised 12 dogs, with a mean age of 5.4 ± 1.8 years and a body weight of 29.7 ± 4.3 kg. Most dogs were Belgian Malinois (58%), German Shepherd (25%), Labrador Retriever (0.8%), and Dutch Shepherd (0.8%) breeds. Dogs were used for drug sniffing (75%), search and rescue (16%), guarding (25%), and explosives detection (16%). Of note, the same dog can perform multiple functions. However, all dogs were submitted to similar daily and weekly workloads. 

### 3.2. Location of TP

All dogs (12) in this sample had at least one trigger point. Overall (i.e., 49 muscle areas palpated on each side of 12 dogs), 72 TPs were detected, with an average of 6 TPs per animal. Of these, 30 (42%) were located on the left side (L) and 42 (58%) on the right side (R). No TPs were detected in the head or neck muscles. With regard to the remaining areas, 32 (44%) TPs were found in spinal muscles, 34 (47%) in hind limb muscles, and 6 (8%) in forelimb muscles (Figure 2). The methods and criteria employed for the palpation of TPs in dogs have proven to be valid, as indicated by the consistent reaction patterns observed during these palpations.

The *longissimus lumborum* had the highest percentage of TPs (42%, area 2, R side), followed *by* the *latissimus dorsi* (33%, region 2, R side), the *quadriceps femoris* (33%, region 2, R side), the *pectineus* and *sartorius* (33% each, R side), and the *gluteus medius* (25%. R and L sides) muscles (Table 1). Figure 3 and Figure 4 show the sites with higher overall TP percentage (i.e., both examiners combined). A table with the findings reported by each examiner is provided in Appendix C. Muscle groups and areas in which no TPs were detected by either of the examiners (0%) were not included in the Table 1.

## 4. Discussion

This study revealed that dogs present TPs just as humans do, but with particular reactions due to palpation. In this population of dogs, the presence of TPs was higher in spinal and hind limb muscles. As in humans [24] and other non-human species [25], the *latissimus dorsi*, *longissimus lumborum*, *gluteus medius*, and *quadriceps femoris* muscles were the most commonly affected. Additionally, it was possible to establish the criteria and reactions resulting from the palpation of TPs in dogs.

Myofascial pain may have different etiologies. Human studies have shown that certain types of work and physical activity increase the incidence of myofascial pain in some muscle groups [26,27]. One study [28] investigating lumbar myofascial pain in patients with no changes in the lumbar spine on magnetic resonance imaging reported a 47.7% prevalence of myofascial pain in the *quadratus lumborum* and implicated TPs as the cause of dysfunction in these patients. Lower back pain associated with lumbar TPs has been described in humans [28,29] and horses [25]. Likewise, TPs in the lumbar region may cause chronic lower back pain in dogs.

These highly trained canines perform duties such as suspect tracking, missing person searches, and narcotics and explosives detection. Their on-duty physical activity entails intense running, pursuit, obstacle traversal, and precise navigation through complex terrains. The presence of TPs in hind limb muscles such as the *gluteus medius* and *quadriceps femoris* in the dogs in this study is justified by the fact these muscles are regularly used in daily activities such as jumping and obstacle clearance [30]. 

In working dogs, hind limb overload may cause micro lesions and fatigue, leading to TP formation [31]. According to Srbely et al. (2010) [32], repetitive muscle contractions are a cause of cell membrane rupture and damage to the sarcoplasmic reticulum, with resultant Ca^2+^ release and breaking of cytoskeleton proteins (desmin, tinine, and dystrophin), which could explain the formation of TPs. These authors also suggested that increased cytochrome C oxidase levels contribute to muscle pain in affected patients. Circulatory deficits may elicit metabolic changes in muscle tissues (the energy crisis hypothesis), with the recruitment of the anaerobic system through pyruvic acid metabolism and increased lactic acid buildup, leading to a drop in muscle pH, reaching 5.0, a value reported in TP vicinity [33]. Aside from pain, this mechanism of injury has profound effects on muscle activation [34] and may negatively impact animal physical performance, as reported in humans [35,36]. 

The higher percentage of right-sided TPs in dogs in this sample may reflect the type of work these animals perform. In training sessions and work shifts, police dogs are submitted to high-intensity exercises and are often positioned on the handler’s left side. Hence, they must constantly turn their head to the right to watch the handler, which could explain the higher percentage of muscle pain on the right side. 

Lower back pain resulting from MPS may impair the performance of working dogs in the long term. In humans, MPS and related back pain are significant absenteeism causes and social isolation, with negative impacts on quality of life [27,37]. Research on German Shepherds revealed that the inability to cope with the physical demands of work is the primary cause of retirement among police dogs, with degenerative diseases and musculoskeletal pain accounting for 65% of retirement requests [38]. 

The trigger points observed in this study seem to be numerous in working dogs, although this remains to be compared with sedentary dogs. Human athletes have also reported myofascial pain induced by physical activities [19,20]. In a study carried out by Gil (2015) [19], at least one TP was detected in each of the fifteen athletes examined. The TPs were thought to be a potential cause of a decreased range of motion in affected athletes. Park et al. (2010) [20] have shown that the location and prevalence of TPs differ significantly between athletes from different sports disciplines, suggesting various athletic activities may lead to muscle injuries at different sites. According to the proposed pathophysiological mechanisms of TP formation [39], frequent micro injuries induce muscle hypercontraction, leading to focal ischemia and inflammation and finally to TP development, which may result from the athletic activity performed [4,6]. 

A single clinical study reporting on myofascial pain in 48 dogs with lameness was found in the Pubmed database [9]. In that study [9], 82 TPs were detected through palpation: 43 TPs located in the *triceps brachii*, 1 in *infraspinatus*, 10 in *peroneus longus*, 8 in *gluteus medius*, 5 in *iliocostalis lumborum*, 12 in *pectineus*, and 3 in *quadriceps femoris.* As in this study, TPs were found in the *quadriceps femoris*, *pectineus*, *gluteus*, and *triceps brachii* muscles, albeit with different prevalences. Etiological differences may explain the prevalence discrepancies between these studies. In the study by Janssens (1991) [9], TPs were assumed to result from past joint/musculoskeletal injuries. In contrast, the dogs in our sample were healthy, and the TPs detected were thought to reflect their daily physical activities, as previously discussed. The remaining publications addressing the topic were review articles, transposing the findings in humans and horses to other species [8,16,25]. 

Assessments in this study were standardized and the examiners were appropriately trained. Nevertheless, the findings underscore the subjective nature of palpation for identifying pain associated with TPs, a major limiting factor for myofascial pain research reproducibility. This is not exclusive to the veterinary field, and similar difficulties have been reported in several articles [5,17,36].

Trigger point recognition criteria used in clinical rehabilitation research between 2007 and 2019 have been examined in a systematic review [5]. Only 129 out of 198 articles provided clear information concerning criteria used for TP identification, 56 failed to provide appropriate descriptions, and only 13 adopted criteria supported by specialists. The most commonly cited criteria were the detection of firm nodules, local contractile reflex, referred pain, and self-reported pain sensation. The first two criteria were also used in this study. 

According to Frank (1999) [17], in a review transposing data from humans, decreased range of motion, palpable taut bands, firm nodules, local pain reaction to palpation, contractile response, weakness in affected muscles, and referred pain are the criteria applicable to dogs. In the present study, we found just pain reaction (as described in the Section 2) and local contractile responses to be applicable to dogs. Evaluating certain parameters, such as weakness sensation, proved to be challenging, and often, the presence of nodules and taut bands were unclear during palpation due to the size of structures and muscles in dogs. Given the subjective nature of these criteria, it is important to emphasize the need for examiner training and the expansion of existing criteria to ensure accurate recognition of canine reaction patterns to TP palpation in future studies.

The findings of this study emphasize the need for preventive measures, ranging from identifying factors causing myofascial pain to adopting therapeutic and preventive physiotherapy interventions [40] to support work-related performance and longevity and possibly provide a good quality of life for these animals. 

Convenience sampling and exploratory research are a major limitation of this study. To estimate prevalence with external validation, a larger sample should have been used, as shown by sample size estimates. Nevertheless, this is the first article to look for TPs in working dogs, and selecting a homogeneous group of dogs (i.e., submitted to the same management, nutrition, and work schedule) led us to exclude other causes for the TPs shown by these dogs. Also, for further research, we recommend using a group of sedentary dogs to understand if the location and percentage of the TPs are different in this control group. 

## 5. Conclusions

Police working dogs show TPs, and the spinal and hind limb muscles are the most commonly affected in the group of this study. The findings of this study may support the implementation of preventive and therapeutic measures in working dogs and can be the basis for further research. The need to establish appropriate criteria for the diagnosis of myofascial pain and to train professionals working in this area to mitigate potential discrepancies inherent in the subjective quantification of pain in animals is emphasized.

## Figures and Tables

**Figure 1 animals-13-02836-f001:**
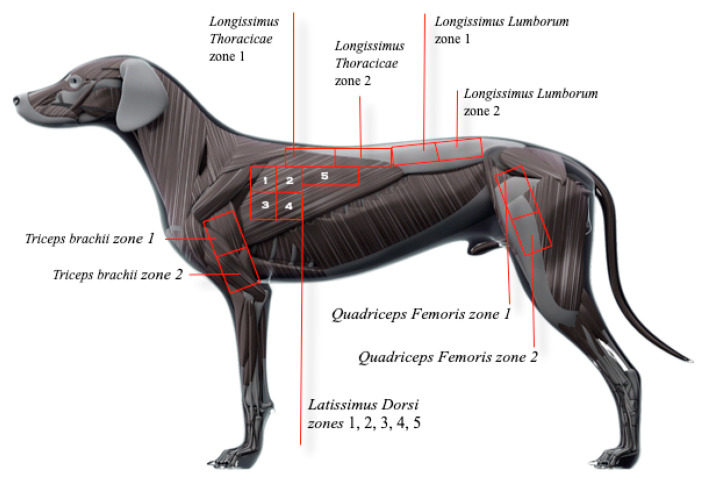
Anatomical references used for subdividing large muscle groups into smaller areas.

**Figure 2 animals-13-02836-f002:**
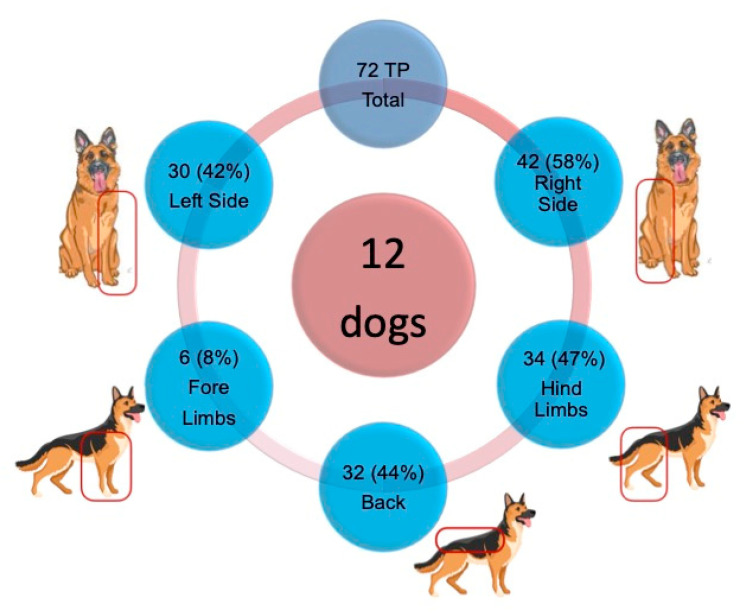
Distribution between sides and main locations of the 72 TPs found in the 12 dogs. R: Right side. L: Left side.

**Figure 3 animals-13-02836-f003:**
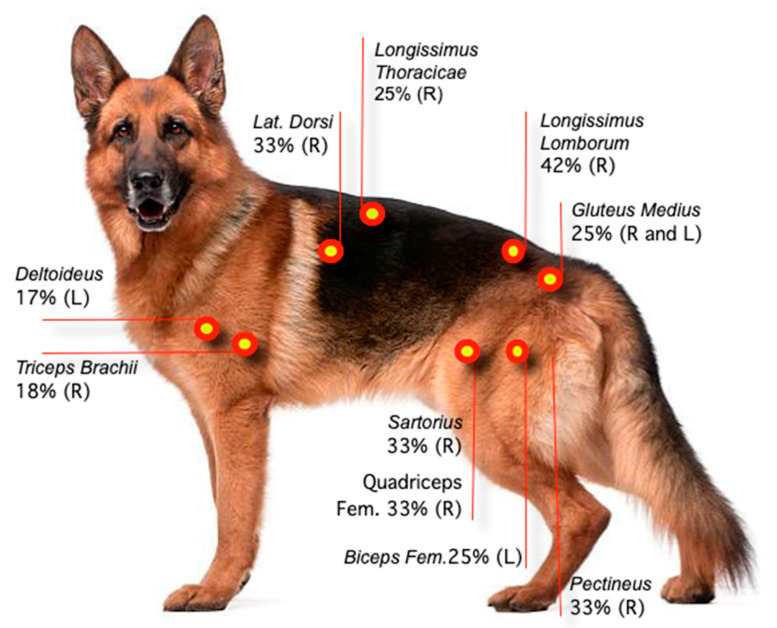
Location and percentage of TPs based on data collected by two examiners. Each muscle assessment was treated as an independent measure. R: Right side. L: Left side.

**Figure 4 animals-13-02836-f004:**
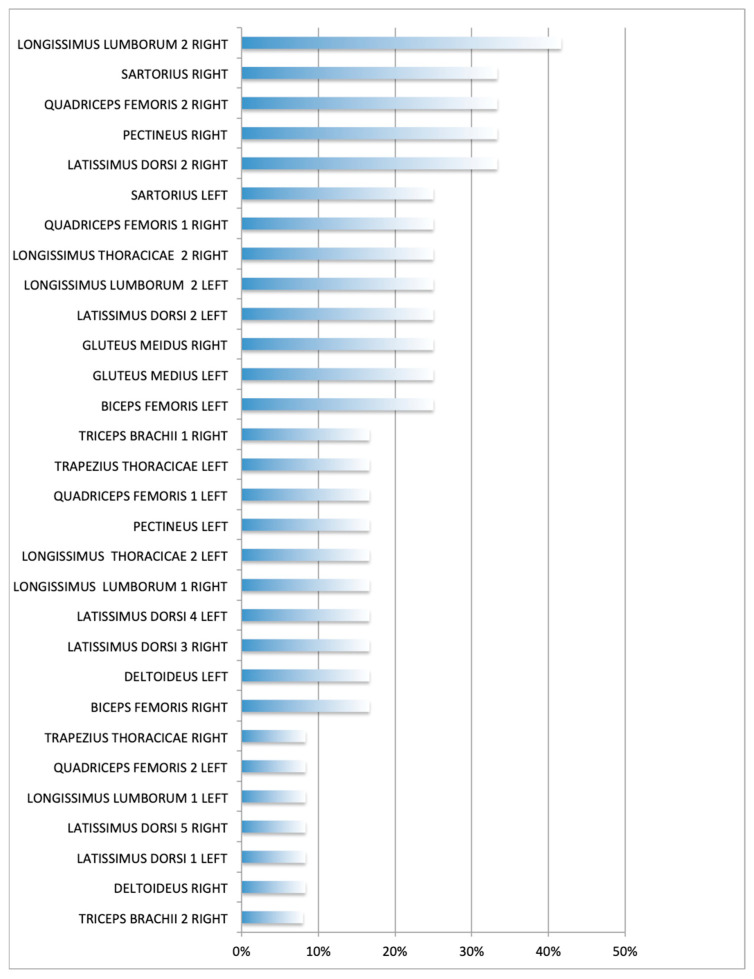
Overall percentage (i.e., both examiners combined) of muscle group/areas affected. Each muscle assessment was treated as an independent measure. Frequency expressed as the No. of muscles affected/No. of muscle groups or areas assessed per side (12 dogs).

**Table 1 animals-13-02836-t001:** Percentage of muscles affected (i.e., both examiners combined) by trigger points in each muscle group or region. Each muscle assessment was treated as an independent measure. Frequency is expressed as the No. of muscles affected/No. of muscle groups or areas assessed per side (12 dogs). Percentage differences between the left and right sides are shown at the bottom of the table.

Muscle Group or Area	Number of Muscle Groups Affected and Percentage	
	Left	Right
Biceps Femoris	3/12 (25%)	2/12 (17%)
Deltoideus	2/12 (17%)	1/12 (8%)
Gluteus Medius	3/12 (25%)	3/12 (25%)
Latissimus Dorsi 1	1/12 (8%)	
Latissimus Dorsi 2	3/12 (25%)	4/12 (33%)
Latissimus Dorsi 3		2/12 (17%)
Latissimus Dorsi 4	2/12 (17%)	
Latissimus Dorsi 5		1/12 (8%)
Longissimus Lumborum 1	1/12 (8%)	2/12 (17%)
Longissimus Lumborum 2	3/12 (25%)	5/12 (42%)
Longissimus Thoracicae 2	2/12 (17%)	3/12 (25%)
Pectineus	2/12 (17%)	4/12 (33%)
Quadriceps Femoris 1	2/12 (17%)	3/12 (25%)
Quadriceps Femoris 2	1/12 (8%)	4/12 (33%)
Sartorius	3/12 (25%)	4/12 (33%)
Trapezius Thoracicae	2/12 (17%)	1/12 (8%)
Triceps Brachii 1		2/12 (17%)
Triceps Brachii 2		1/12 (18%)
Percentage difference between L and R sides	30/72 (42%)	42/72 (58%)

## Data Availability

Not applicable.

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
