# Peer review of "Location of Trigger Points in a Group of Police Working Dogs: A Preliminary Study"

_animals, 2023, doi:10.3390/ani13182836_

Round 1

Reviewer 1 Report

Thank you for the opportunity to review the manuscript “Location of Trigger Points in a Group of Police Working Dogs: A Preliminary Study. The article is well-written and is of interest, especially for those practicing in integrative medicine and doing myofascial research. My comments are below.

Please review your muscle names. There are multiple mistakes throughout the text with improper (check the NAV when in doubt) or confusing terms:

               Line 131: Sternocephalicus and sternomastoideus are both written. The sternomastoideus is one of the heads of the sternocephalicus.

               Line 133: supraescapularis, infraescapularis- there should be no “e”; brachioradiali should have an “s” at the end;

Line 134: externsor carpi, digitorum communis- do you mean extensor digitorum communis?

Line 137-138 and elsewhere, including figures: there is no “dorsi” in the longissimus muscle group. It is simply longissimus thoracis, etc.

Line 138: serratil thoracicae- ? serratus ventralis thoracis?

Throughout- “gluteus”- you are very specific with most muscles, but not this one. Please be more specific.

Some of the palpated muscles with trigger points are just deep to other muscles with trigger points (e.g., longissimus thoracis area 1 or 2 lying deep to latissimus dorsi). Is there any assurance/explanation that the issue is actually in the deeper muscles and not the ones superficial to them (or indeed, even within the more superficial fascia)?

Line 249: While jumping, and perhaps obstacle clearance, are reasonable explanations, walking is a weak one. Many muscles are used during walking and this, in and of itself, does not explain the cause of a trigger point.

Appendix A: Description of longissimus groups as paravertebral- the transversospinalis groups is truly paravertebral; longissimus is just lateral to that group. Please clarify.

The manuscript is well written and the English needs only minor editing. The muscle names, however, are not consistent and the reviewer recommends consulting the NAV (Nomina Anatomica Veterinaria) for consistency and correctness in anatomical terminology.

Author Response

Authors: Dear Reviewer,

We sincerely appreciate the time and effort you have dedicated to review our article. Your comments and valuable feedback have been helpful in improving the quality and clarity of our work. We made the appropriate changes as suggested, indicated by red markings in the manuscript.

               Line 131: Sternocephalicus and sternomastoideus are both written. The sternomastoideus is one of the heads of the sternocephalicus.

Authors: Corrected. We've removed "sternomastoideus".

               Line 133: supraescapularis, infraescapularis- there should be no "e"; brachioradiali should have an "s" at the end;

Authors: Corrected –  supraspinatus, infraspinatus instead. Corrected brachioradialis

Line 134: extensor carpi, digitorum communis- do you mean extensor digitorum communis?

Authors: Corrected. "extensor carpi" was written twice, removed one. Corrected "extensor digitorum communis"

Line 137-138 and elsewhere, including figures: there is no "dorsi" in the longissimus muscle group. It is simply longissimus thoracis, etc.

Authors: Corrected.

Line 138: serratil thoracicae- ? serratus ventralis thoracis?

Authors: Corrected - serratus ventralis thoracis

Throughout- "gluteus"- you are very specific with most muscles, but not this one. Please be more specific.

Authors: Yes, I have rechecked the anatomical figures used by the evaluators to verify where they marked the TP in the anatomical reference, whether it was in the gluteus superficialis or M. gluteus medius. They palpated both muscles, but the TPs were found in the gluteus medius. Corrections were made in all figures, tables, etc., accordingly. As suggested, we reviewed and revised all the points and muscles using the "NOMINA ANATOMICA VETERINARIA," 6th edition (2017). We also corrected all other names in pictures, graphics, and tables.

Some of the palpated muscles with trigger points are just deep to other muscles with trigger points (e.g., longissimus thoracis area 1 or 2 lying deep to latissimus dorsi). Is there any assurance/explanation that the issue is actually in the deeper muscles and not the ones superficial to them (or indeed, even within the more superficial fascia)?

Authors: It is a common question in myofascial pain palpation. To effectively palpate myofascial pain, it is advisable to begin with gentle pressure on the superficial layers. If no pain is detected, the pressure is gradually increased to explore deeper layers where pain can be identified. Developing sensitivity through training and experience is essential for this process.

If there is already pain in the superficial layers, it might be challenging to access the deeper muscles. A parallel can be drawn with palpation of the abdomen. When palpating softly, you can feel the abdominal skin and superficial fascia, but you can reach the muscles with increased pressure. Going even deeper allows you to palpate specific organs, using anatomical references to understand their location, such as the bladder or stomach.

However, it's essential to acknowledge that this technique involves some subjectivity, as discussed in the article. This subjectivity is inherent to the palpation method, whether it pertains to joints, bony structures, or muscles.

Line 249: While jumping, and perhaps obstacle clearance, are reasonable explanations, walking is a weak one. Many muscles are used during walking, and this, in and of itself, does not explain the cause of a trigger point.

Authors: Yes, they engage in extensive physical activities regularly. These activities include jumping, walking long distances, climbing stairs and hills, and overcoming various obstacles. These rigorous activities are integral to both their training and daily work routines. We have expanded the sentence to provide a more comprehensive explanation. Thank you for your understanding.

Appendix A: Description of longissimus groups as paravertebral- the transversospinalis groups is truly paravertebral; longissimus is just lateral to that group. Please clarify.

Authors: Corrected. We changed the description to "dorsal". Loginssimus is one of the main epaxial muscle, the easiest to palpate in this area.

Once again, thank you for your contribution to our manuscript. Your input has been immensely beneficial, and we are honoured to have had the opportunity to receive such feedback from you.

Reviewer 2 Report

This is a very interesting work, well written, clear and ordered.

There are a few comments I would like to make and some doubts I have after reading your work.

You say that palpation for identifying muscular pain has a subjective nature, and this is not exclusive to the veterinary field. On materials and methods you also say that the dogs were palpated in alternating turns by two observers. How many times did each observer examined every dog? It would be good to know the intraobserver variability of the method.

Also, reviewing the results of the two observers, it seems that their results are slightly different. Did you compared their results statistically? This information would add value to the relevance of your  study.

At the Materials and Methods section you mention that the battalion military veterinarians were trained to stretch the dogs’ muscles and treat them for muscular pain. Did you check the dogs after the treatment or have you considered palpating them after the treatment to compare the method, and to evaluated the efficacy of the treatment?

I agree with your suggestion that it would be ideal to have a control group to compare the results with this studied population.

This work could be the opening door to more studies on a barely explored field on small animal medicine.

Thank for your time and effort.

Author Response

This is a very interesting work, well written, clear and ordered.

There are a few comments I would like to make and some doubts I have after reading your work.

You say that palpation for identifying muscular pain has a subjective nature, and this is not exclusive to the veterinary field. On materials and methods you also say that the dogs were palpated in alternating turns by two observers. How many times did each observer examined every dog? It would be good to know the intraobserver variability of the method.

Also, reviewing the results of the two observers, it seems that their results are slightly different. Did you compare their results statistically? This information would add value to the relevance of your study.

Authors: Dear Reviewer

Thank you very much for taking the time to contribute to our work. Your comments were of great value, not only for this article but also for our future papers. Your expertise has enriched the quality of our research, and we truly appreciate your valuable input.

This police dog data is our preliminary "phase one" paper. We decided to publish our initial findings as a "communication" to lay the groundwork for future papers. Since there is limited published research on this topic, we wanted to demonstrate that dogs also suffer from TP, how to identify it through palpation, and the related reactions to it.

Reviewer:

On materials and methods you also say that the dogs were palpated in alternating turns by two observers. How many times did each observer examined every dog?

Authors: During the evaluation of the dogs, we made sure to palpate each one once, and the assessment order was randomized to avoid bias. We added the following sentence in the text, marked in blue:

One evaluator palpated all regions of the animal at once following the order described below, after the other evaluator examined the same animal. In order not to create bias, the order of the evaluator's examination was alternated.

We also excluded in alternating times in the previous sentence.

Reviewer:

It would be good to know the intraobserver variability of the method.

Also, reviewing the results of the two observers, it seems that their results are slightly different. Did you compare their results statistically?

Authors:  Indeed, the difference in the results between the evaluators is considerable, but it is similar to what has been observed in human studies.

And, Yes we conducted an intra-observer evaluation in this small group of dogs. Initially, we had planned to include the statistical results in this article.

You can find below some of the data that was not included in the version of the manuscript:

-The total number of muscles palpated was 41. However, when considering subdivisions of larger muscle groups, we palpated 49 muscle regions with 98 assessments per dog (including both sides). 12 dogs x 98 palpations per dog = 1176 each evaluator.

-Each evaluator conducted 1176 assessments.

- The agreement between evaluators (kappa = 0.32,agreement 95.15 %)

 (This result is comparable to some results obtained in human studies).

Table 1: Presence of pain trigger points (TP) according to evaluator 1 (line) and to evaluator 2 (column) in each assessment (kappa = 0.32, agreement 95.15 %).

No TP

Yes TP

Total

No TP

1104 (93.88%)

32 (2.72%)

1136 (96.6%)

Yes TP

25 (2.13%)

15 (1.28%)

40 (3.4%)

Total

1129 (96%)

47 (4%)

1176 (100%)

We expanded this analysis of the correlation for the DIVAS each evaluator marked for each point, and reached: Divas: kappa = 0.21,p = 0,agreement = 94.3 %.

Note that the agreement is also much higher in detecting that there is NO myofascial pain/TP. We observed a similar result in phase 2, where 35 dogs with a specific disease were included. Clearly, establishing the patterns of myofascial pain in dogs is necessary before we can develop a better method for identifying the dogs' reactions. Training is also essential, as for any pain palpation. The same discussion is present in humans' articles, with a lot of research trying to establish methods and increase agreement, as palpation keeps being the main diagnostic method for TPs - and they have an advantage: humans tell them where or when there is pain. So I believe our path in veterinary medicine will be much more full of difficulties.

As you noticed, it is a whole new objective, analysis, discussion and conclusion, that’s the reason why we decided to expand the correlation to another paper together with the results of phase 2. With this first article on police dogs, the idea is to lay the foundation for future publications. We will proceed by comparing evaluators, establishing prevalence, and, towards the end of our study, comparing treatments.

Still, we were unsure if this observer analysis would offer any assistance in this article to help vets understand the complexity of the subject or if it would only enhance the doubts about it. What do you suggest? 

We thought it would be interesting to share these information with you so can have more material to enhance the comprehension of our data. If you have any further thoughts or suggestions, please feel free to share them. We greatly appreciate your contribution and look forward to receiving any other comments.

Reviewer:

At the Materials and Methods section you mention that the battalion military veterinarians were trained to stretch the dogs' muscles and treat them for muscular pain. Did you check the dogs after the treatment or have you considered palpating them after the treatment to compare the method, and to evaluated the efficacy of the treatment?

Authors: Unfortunately, too few dogs were available for further evaluation. Some of them retired, some were transferred to other battalions, and others were part of a reproduction program. While we were aware of this limitation, our strategy was to teach the people responsible for the dogs these techniques, to assist them and future dogs under their care. The treatment phase constitutes the third part of our new study, and we are currently collecting data for this phase.

I agree with your suggestion that it would be ideal to have a control group to compare the results with this studied population.

This work could be the opening door to more studies on a barely explored field of small animal medicine.

Authors: Once again, thank you for your thoughtful and constructive comments; they will undoubtedly aid us in improving the overall quality of our work.